# Pregnancy in Patients with Pulmonary Arterial Hypertension in Light of New ESC Guidelines on Pulmonary Hypertension

**DOI:** 10.3390/ijerph20054625

**Published:** 2023-03-06

**Authors:** Karolina Barańska-Pawełczak, Celina Wojciechowska, Wojciech Jacheć

**Affiliations:** 1Department of Cardiology, Specialistic Hospital in Zabrze, 41-800 Zabrze, Poland; 2Second Department of Cardiology, Faculty of Medical Sciences in Zabrze, Medical University of Silesia, 40-055 Katowice, Poland

**Keywords:** pregnancy, pregnancy outcome, high-risk pregnancy, pulmonary arterial hypertension, right heart failure, right ventricular dysfunction, adoption

## Abstract

Pulmonary arterial hypertension (PAH) is defined as an elevated mean pulmonary artery pressure (mPAP) of >20 mmHg together with a pulmonary arterial wedge pressure (PAWP) of ≤15 mmHg and pulmonary vascular resistance (PVR) of>2 Wood units (WU). Although the total mortality of pregnant women with PAH has decreased significantly in recent years and is reported to be around 12% in some databases, total mortality is still at an unacceptably high percentage. Moreover, some subgroups, such as patients with Eisenmenger’s syndrome, have a particularly high mortality rate of up to 36%. Pregnancy in patients with PAH is contraindicated; its appearance is an indication for a planned termination. Education of patients with PAH, including counseling on effective contraception, is essential. During pregnancy, blood volume, heart rate, and cardiac output increase, while PVR and systemic vascular resistance decrease. The hemostatic balance is shifted towards hypercoagulability. Among PAH-specific drugs, the use of inhaled or intravenous prostacyclins, phosphodiesterase inhibitors, and calcium channel blockers (in patients with preserved vasoreactivity) is acceptable. Endothelin receptor antagonists and riociguat are contraindicated. Childbirth can take place through either vaginal delivery or caesarean section; similarly, neuraxial and general anesthesia have proven indications. In a situation where all pharmacological options have been used in pregnant or postpartum patients in a serious condition, veno-arterial ECMO is a useful therapeutic option. For PAH patients who want to become mothers, an option that does not endanger their lives is adoption.

## 1. Introduction

According to the 2022 European Society of Cardiology/European Respiratory Society (ESC/ERS) guidelines for the diagnosis and treatment of pulmonary hypertension (PH), it is defined as an elevated mean pulmonary artery pressure (mPAP) of >20 mmHg, and the definition of pulmonary arterial hypertension (PAH) also implies an pulmonary arterial wedge pressure (PAWP) of ≤15 mmHg and pulmonary vascular resistance (PVR) of >2 Wood units (WU). Anew clinical classification of PH has also been presented (Figure 1) [1]. PAH also belongs to group IV according to the modified World Health Organization (mWHO) classification of maternal cardiovascular risk in that it is related to an extremely high risk of maternal mortality or severe morbidity [2]. Despite significant improvements in the treatment of patients with PAH, pregnancy is still regarded as a contraindication not only due to high maternal morbidity and mortality but also due to serious fetal consequences. Historical data determined maternal mortality in PH to be 30% in primary pulmonary hypertension, 36% in Eisenmenger’s syndrome, and 56% in secondary vascular PH (in a heterogeneous group of patients with, e.g., PH connected with the use of incriminated drugs, some cases of congenital origin or the result of thromboembolism, hepatitis, or systemic connective tissue or vascular inflammatory disease), and neonatal mortality from 11% to 13%, similar to the rates as mentioned in the above groups [3]. Current data, based on a review of available PAH patient databases from 2008 to 2018, show that mortality for pregnant PAH patients remains high, although not as high as in previous decades. The total maternal mortality rate is 12%:20% in idiopathic PAH, 11% in PAH associated with congenital heart disease, and 6% in PAH associated with other conditions, while the overall neonatal mortality rate is about 4%, although some data still show particularly high mortality in Eisenmengear syndrome (36%), which additionally represents a contraindication for pregnancy in such patients. 

The main causes of death, according to the order of occurrence, are right ventricle failure, cardiac arrest, pulmonary hypertension crisis, preeclampsia, and sepsis [4,5].

There are still situations where it is necessary to manage pregnancy in patients with PAH or even the diagnosis of PAH during pregnancy. These circumstances require a well-planned, multidisciplinary approach with detailed monitoring during gestation and after delivery. The aim of this review is to present the current state of knowledge on pregnancy in patients with PAH, and additionally, we would like to share the experience of our center regarding the issue of motherhood in such patients.

## 2. Risk Factors

In the case of PAH patients, not only during pregnancy, but it is also optimal to achieve and maintain the lowest possible risk profile. In accordance with the current ESC recommendations, the determinants of good PAH control include no clinical signs of right heart failure, no progression of symptoms, no syncope, a I or II WHO functional class, and 6min walking distance of >440 m. Among the parameters of cardiopulmonary exercise testing, the low risk of death is associated with peak oxygen consumption (VO_2_) of >15 mL/min/kg, ventilatory equivalents for carbon dioxide (VE/VCO_2_ slope) of <36, and among the laboratory parameters, with the concentration of brain natriuretic peptide (BNP) of <50 ng/L or N-terminal pro-brain natriuretic peptide (NT-proBNP) of <300 ng/L. The echocardiographic parameters of a good prognosis are the right atrium (RA) area of <18 cm^2^, no pericardial effusion, tricuspid annular plane systolic excursion/systolic pulmonary arterial pressure index (TAPSE/sPAP) of >0.32 mm/mmHg, and among the MRI parameters, belong the right ventricular ejection fraction (RVEF) of >54%, SVI > 40 mL/m^2^ and right ventricular end-systolic volume index (RVESVI) of >42 mL/m^2^. The hemodynamic markers of low risk of death are right atrial pressure (RAP) of <8 mmHg, cardiac index (CI) of ≥2.5 L/min/m^2^, mixed venous oxygen saturation (SvO_2_) of >65%, and stroke volume index (SVI) of >38 mL/m^2^ (1). In addition to generally recognized risk factors, a woman’s first pregnancy, general anesthesia, and elevated PVR (but without a clear cut-off point) have been identified as risk factors for the mother [6,7]. In addition, sPAP has been established as an independent risk factor, with a cut-off value of 56 mmHg [8]. Based on the observation of a group of 249 pregnant patients (214 with PAH and 35 patients with PH related to left heart disease), PH before pregnancy, delivery at ≥28 weeks gestation, and severe pulmonary hypertension with PAP of >80 mmHg are considered as independent risk factors for cardiac complications by multivariate analysis [9].

Pregnancy in PAH patients is a risk not only for the mother but also for the baby. In a group of 260 pregnant patients with PAH with congenital heart disease, 10 deaths among their offspring were reported, mainly due to prematurity and the resulting low birth weight and immaturity of the offspring. In the whole study group, a high percentage of premature deliveries (83.7%) and low birth weight (83.7%) were observed; however, in most cases, it did not affect the development of the newborns as compared to their peers [10].

In the case of pregnancy in patients with PAH, it should also be remembered that children may inherit mutations that increase the risk of PAH. Mutations responsible for the development of the disease have been found in idiopathic and familial PAH, anorexigen-associated PAH, pulmonary veno-occlusive disease pulmonary capillary hemangiomatosis (PVOD/PCH), and PAH associated with congenital heart disease. If a mutation in PAH is identified, it should be classified as hereditary PAH (HPAH), and the available data show that around 25–30% of IPAH cases should be classified into this group [11].

## 3. Birth Control

Recommendations against becoming pregnant may lead to the disruption in the emotional well-being of patients with PAH, with the recommendation being a source of frustration resulting from limitations that cannot be overcome [12]. More patients with PAH are considering having children. Improving the survival rate of patients with pulmonary arterial hypertension during pregnancy may lead to taking the risk of becoming intentionally pregnant or maintaining pregnancy in the event of conception. Adequate counseling is essential not only to provide guidance on the maternal risks associated with pregnancy but also to provide guidance about the teratogenic properties of specific drugs used in PAH.

Effective contraception should be used by all PAH patients of childbearing age. Based on the Centers for Disease Control and Prevention recommendations on the effectiveness of different methods of contraception, tier I methods (male and female sterilization, implants, and intrauterine devices) with failure rates of less than 1% per year are recommended for all patients for whom pregnancy is contraindicated (WHO pregnancy risk group IV) and they can be used as a single form of contraception. Tier II methods (pills, injectables, rings, patches, and diaphragms) with failure rates of 6–12% per year do not have such recommendations; therefore, two methods should be used simultaneously, e.g., hormonal and barrier methods or two barrier methods [13,14,15]. It should be remembered that although combined oral contraceptive pills are a safe method of contraception, the estrogen component could be associated with an increased risk of thromboembolic complications. Progestogen-only contraceptive methods could be an alternative, due to the lack of an increased risk of thrombosis; however, they are contraindicated for women receiving bosentan for pulmonary arterial hypertension because it induces the p450 enzyme and, therefore, can reduce the effectiveness of progestogen-only pills [16].

## 4. Termination of Pregnancy

Patients with PAH who become pregnant should receive comprehensive counseling regarding their condition, pregnancy-related risks to the mother and fetus, and treatment options, including the termination of pregnancy. If the termination of pregnancy is an accepted therapeutic option, it should be performed as early as possible. Taking into account all methods of termination of pregnancy in patients with pulmonary hypertension (PH-1 and PH-2), i.e., laparoscopic, open, or transvaginal, the total periprocedural mortality rate is estimated at 4–6% [17]. Data available in the literature support the safety of planned termination of pregnancy. In a multicenter pregnancy registry in PAH patients, out of eight abortions—two spontaneous and six planned—all of the induced termination patients underwent this procedure without complications, while both women with spontaneous abortions died [7]. Late termination of pregnancy, which is additionally more often performed as a surgical procedure, is associated with an increased risk of complications, due not only to a longer period of stress on the circulatory system with a longer duration of pregnancy but also due to the use of anesthesia [18].

## 5. Hemodynamics Changes in Pregnancy

The growing uterus and its metabolic requirements as well as fluctuations in hormone levels that appear at the onset of pregnancy and last until the postpartum period affect the function of the entire body. During pregnancy, blood volume increases by about 50% compared to the period before pregnancy, which increases the burden on the circulatory system and also causes a decrease in the blood concentration of the drugs used so far. The red blood cell (RBC) volume increases, but not to the same extent as the plasma volume, which can lead to dilutional anemia [19]. Heart rate (HR) increases during pregnancy as well as the left ventricle end-diastolic dimension and consequently stroke volume (SV) and CI [20]. An increase in cardiac output (CO) is seen, mainly due to the increase in HR rather than the increase in SV [21], as well as a significant decrease in PVR and systemic vascular resistance (SVR) [22]. Peak CO is usually achieved at the beginning of the third trimester [23]. Among women with PAH, the increase in sPAP during pregnancy has been proven, and although it decreases after termination, its value is still higher than at the beginning of pregnancy, which may suggest that pregnancy accelerates the progression of the disease [24]. 

An important element influencing an individual’s health condition in pregnancy is also an increase in the tendency to suffer from supraventricular and ventricular arrhythmias [25]. Blood pressure (BP) data are less consistent, with no difference in blood BP values throughout pregnancy [26] as well as there being a drop in BP in the second trimester followed by an increase in the third trimester [27].

In patients with a normal pregnancy, in order to reduce the risk of hemorrhagic complications during childbirth, the hemostatic balance is shifted towards hypercoagulability. There is a decrease in the concentration of free protein S—a blood coagulation inhibitor—as well as acquired activated protein C resistance, and an increase in coagulation factors and fibrinogen. Together, they may lead to an increased risk of thromboembolic complications. The hemostatic changes in pregnancy usually normalize within 4 to 6 weeks after delivery [28]. Changes have been summarized in Figure 2.

The physiologic cardiovascular adaptations of pregnancy and labor appear in the vast majority of women without circulatory decompensation, but increased plasma volume and increased CO often cannot be handled by a failed right ventricle which may lead to right ventricular failure in patients with PAH.

## 6. Hemodynamics Changes during Delivery

The risk of cardiovascular decompensation for patients with PAH is particularly high in the perinatal period, which is associated with rapid hemodynamic changes that occur during this period. It is influenced by many factors, including uterine contractions, pain, anxiety, bleeding, or anesthesia. During labor, each uterine contraction causes the transfer of about 500 mL of blood into the mother’s bloodstream, which increases CO and systemic pressure [29]. Cardiac output rises by 15% in early labor and up to 50% in the final stage. In the early postpartum period, CO may still increase, even up to 80%, which is due to increasing SV by the involution of the uterus and the reduction in peripheral edema [30]. In addition, a sudden drop in progesterone and estrogen levels after the delivery of the placenta causes an increase in vascular resistance [31]. Cardiac output and HR return to pre-pregnancy values within 6 weeks, but in most cases, recovery is much faster. Stroke volume also decreases during the postpartum period, but a further reduction is seen up to six months after delivery [32].

## 7. Pharmacotherapy during Pregnancy

The standard medications for the treatment of PAH are calcium channel blockers (nifedipine, diltiazem, and amlodipine—for patients with preserved vasoreactivity), endothelin receptor antagonists (ambrisentan, bosentan, and macitentan), phosphodiesterase type 5 inhibitors (sildenafil and tadalafil), guanylate cyclase stimulators (riociguat), prostacyclin analogues (beraprost, epoprostenol, iloprost, and treprostinil), and prostacyclin receptor agonists (selexipag). Based on the Food and Drug Administration recommendations for the use of the above-mentioned drugs in pregnancy, Table 1 provides a summary [33,34,35,36,37,38,39,40,41,42,43,44,45,46,47].

Endothelin receptor antagonists (ambrisentan, bosentan, and macitentan) and guanylate cyclase stimulators (riociguat) are contraindicated in pregnancy due to their teratogenic properties. Prostacyclins (epoprostenol, treprostinil, and iloprost) and phosphodiesterase inhibitors (sildenafil and tadalafil) are among the basic medications recommended in pregnancy for patients with PAH and have been proven effective in reducing mortality. Responders (patients with preserved vasoreactivity) may benefit from the use of calcium channel blockers. Another therapeutic option is also inhaled NO, which, through its local action, improves ventilation/perfusion mismatch and oxygenation without reducing system pressure [48]. Symptoms of increasing right ventricular failure in pregnant women with PAH usually begin to appear in the second trimester; therefore, due to increasing symptoms, some centers practice scheduled admission to hospital at around 30–32 weeks of pregnancy to start treatment with intravenous prostanoid infusions [49].

Due to the hypercoagulability state in pregnancy and possible thromboembolic complications, which in patients with PAH may additionally strain their already low cardiovascular reserves, the use of low-molecular-weight heparins is recommended [50].

Data on breastfeeding in patients using PAH therapy are limited; there are no clear recommendations, although there are reports in which there are patients undergoing PAH-targeted therapy with PDE5 inhibitors or parenteral prostacyclins breastfeeding without any known complications [51]. The concentration of sildenafil in breast milk varies from1.64 to 4.49 ng/mL, and that of its major metabolite, N-desmethylsildenafil, varies from 1.18 to 1.82 ng/mL [52].

## 8. Delivery and Puerperium

Pregnant patients with PAH should be under the care of an experienced, multidisciplinary team and undergo regular check-ups—initially at least once a month and even weekly in the last trimester [53]. The optimal time and type of delivery among PAH patients have not been determined, but it is recommended that it should be scheduled in order to avoid situations where inexperienced staff will be required to look after the patient in labor. During labor, close monitoring of HR, saturation, and BP is required, as well as, in some situations, central venous pressure and invasive arterial pressure monitoring and a form of CO monitoring. There are no clear benefits for using a pulmonary artery catheter, and it may increase the risk of damage to the pulmonary artery or cause thrombotic complications [54].

Natural childbirth entails a lower risk of complications—lower blood loss and thus lower hemodynamic fluctuations, as well as a lower risk of thromboembolic complications and local infection; on the other hand, prolonged labor puts additional strain on the already small capacity reserves in patients with PAH and carries risks associated with the Valsalva maneuver during labor and vasovagal response, resulting in the risk of decreased venous return [55,56,57]. The dominant method of delivery is cesarean section, although there are no robust data supporting this model [53,58]. Planned cesarean section is most often performed between 32 and 36 weeks of pregnancy, when the baby is developed enough and to avoid excessive stress on the mother’s health in the final stage of pregnancy [50]. Additionally, the risk of converting vaginal delivery to a cesarean section can be avoided, which can happen in the case of hemodynamic instability or perinatal complications [29]. Routine use of oxytocin should be avoided as it increases PVR which may aggravate the symptoms of PAH. Oxytocin is administered in a slow infusion to reduce the effect on the circulatory system [59,60].

Regional neuraxial anesthesia rather than general anesthesia is recommended and selected during labor. Spinal epidural or epidural anesthesia with a slow titration of an agent is usually recommended because single bolus epidural anesthesia can cause hypotension [61]. Data on differences in mortality in patients with PAH depending on the type of anesthesia used are inconsistent; however, in some studies, there were no statistically significant differences in maternal and neonatal mortality between subarachnoid, epidural, and general anesthesia, but changes in cardiovascular hemodynamics were much more noticeable in the case of general anesthesia, which led to an extension of the time of mechanical ventilation and the length of stay in intensive cares unit and hospitalization. General anesthesia is usually chosen for particularly high-risk patients, such as urgent deliveries related to fetal or maternal decompensation or contraindications to neuraxial anesthesia, which may lead to increased perioperative mortality. In addition, anesthetic agents used in general anesthesia reduce myocardial contractility and intubation, and positive pressure ventilation increases PVR [1,62].

The greatest risk of death for mothers is within 30 days postpartum and is associated with RV failure [3].PAH patients after delivery require close monitoring, with particular emphasis on avoiding fluid overload and right ventricular decompensation and maintaining adequate systemic pressure. Very often intensive care unit treatment is required, and it should be adapted to the variability of PAH. After delivery, all PAH-specific drugs can be used, regardless of their teratogenic properties. Excessive diuresis should be avoided as it can lead to an increase in PVR; it is recommended to use medications that improve cardiac contractility (such as milrinone, dobutamine, and levosimendan) and that do not increase the PVR:SVR ratio [31,63,64]. In patients with persistent cardiogenic shock, norepinephrine or/with vasopressin may be used, with an emphasis on vasopressin which in low doses does not affect the constriction in pulmonary circulation, which is especially important in pulmonary hypertension [65,66].

Patients in whom the above-mentioned pharmacotherapy is ineffective are candidates for mechanical circulatory support in the form of an intra-aortic balloon pump (IABP), right ventricular assist device (RVAD), or extracorporeal membrane oxygenation (ECMO).The first two solutions may be beneficial in a situation where the dominant problem is hypotension and circulatory failure, with good blood oxygenation maintained. However, in the case of patients with PAH, we usually deal with low blood saturation; hence, the use of ECMO is much more beneficial [67]. Veno-arterial ECMO, as a bridge to transplantation or recovery, seems to be a particularly useful therapeutic option as it can completely replace the heart function by unloading the right ventricle as well as maintaining adequate peripheral perfusion and replacing the work of the lungs by oxygenating the blood [68]. Although there are reports of cases in which, thanks to ECMO, it is possible to lead PAH patients out of acute cardio-respiratory failure, so that they are able to leave the hospital, the use of ECMO applies to a group of patients in a particularly serious condition, which translates into very high mortality [58,69,70]. Regardless of the use of ECMO, atrial septostomy remains a therapeutic option to relieve the right ventricle and improve left ventricle filling; however, it exacerbates blood hypoxemia [71]. The final procedure is lung transplantation. There is a description of a woman who underwent such surgery 11 days after delivery and was discharged almost 3 months later, but the availability of donors remains the main limitation [72].

## 9. Our Center’s Recommendations for Motherhood

Among PAH patients under the care of the 2nd Department of Cardiology, Faculty of Medical Sciences in Zabrze, Silesian Medical University in Katowice, all pregnancies occurred before the diagnosis of PAH was made. The diagnosis of PAH is a particularly difficult period for patients in which they have to face the consciousness of being ill with an incurable, progressive disease and change many life plans, including those regarding motherhood. It is important to educate patients, including emphasizing the contraindications of pregnancy in PAH and the fact that even a well-controlled disease can have an unpredictable course. The solution for patients who want to be mothers is adoption, and two of our patients chose this option. 

Case 1. 

Patient 1, born in 1983, with PH after correction of the common atrium by age 5. The diagnosis of PH was confirmed in 2006 during right heart catheterization (RHC) with an inhaled NO (iNO) reversibility test. The patient’s pulmonary artery (PA) pressure was 65.6/28.2 (mean 40.7 mmHg) and 58.8/29.8 (mean 39.5 mmHg) during NO inhalation and their PVR was 584.3 and 496.6 dyna, respectively. The patient was treated with sildenafil (20mg three times a day) from 2008, and in 2021, the therapy was extended with the addition of ARB—bosentan, after RHC where the patient’s PA pressure was 109/39 mmHg (mean 62 mmHg) and their PVR was 11.5 WU. The first NT-proBNP concentration determined in 2008 was 182 pg/mL, with small fluctuations and progression over the years to 471.30 pg/mL in 2022. The patient remains in WHO functional class II; however, extending therapy to include prostanoids is currently being discussed.

Case 2.

Patient 2, born in 1980, with IPAH diagnosed in 1997 and confirmed during RHC with an iNO reversibility test in 2003. The patient’s PA pressure was 90.0/48.0 (mean 62 mmHg) with a reduction to 50.0/28.4 (mean 35.3 mmHg) during NO inhalation, and following pretreatment with sildenafil, their PVR was 1607 and 905.4 dyna, respectively. Sildenafil was introduced as the permanent treatment and its beneficial hemodynamic effect was confirmed after 4 weeks; the value of PA pressure in RHC was 67.5/28.4 (mean 38.4 mmHg) and the PVR was 736 dyna. The patient has remained clinically stable over the years, with the measurement values obtained in the RHC at a comparable level, stable WHO functional class I/II, and NT-proBNP concentration levels below <200 pg/mL. Because of a continuously persistent vasodilatation reserve during NO inhalation, therapy of PAH was switched from sildenafil to riociguat (2.5 mg three times a day) which allowed for additional reduction in mPAP from 40.5 mmHg in 2020 to 31 mmHg in 2020 and a reduction in PVR from 6.2 WU to 4.05 WU. 

In accordance with Polish law, non-incapacitated adults who have appropriate moral and health capacities and a stable financial situation as well as having obtained the qualified opinion of an adoption center and a certificate of completion of training organized by the center may apply for adoption. Current PAH therapy, which in Poland is fully financed by the government, allows good disease control in most cases, which results in maintaining daily life activity and in some cases, even the opportunity to complete light professional work. The examples of PAH patients presented above who have adopted children concern patients with a stable form of the disease, with little progression over the years of follow-up and good control, giving a chance of survival that will allow them to raise their children. Only in this situation can adoption be considered. It is a difficult and complex problem; on the one hand, it provides PAH patients with the opportunity to become mothers without exposing them to the risks associated with pregnancy, improves their mental condition, and provides abandoned children or orphans with the chance to grow up outside of an orphanage in a loving family; on the other hand, it is the institution’s responsibility to prevent situations in which the adopted children may lose their foster parent and family again.

## 10. Conclusions

Based on data from available registries, the improvement in maternal and infant prognosis among pregnant patients with PAH is significant, but total mortality rates are still unacceptably high. Rapid diagnosis of PAH is essential to avoid situations in which the disease is diagnosed during pregnancy, as well as adequate education of patients regarding having children so that it is possible for all decisions to be made with full awareness. Case reports on successful pregnancies and deliveries have been frequently published in the literature, which may give the impression of a good prognosis. Successful completion of pregnancy in PAH patients does not mean that after giving birth, the patient’s condition will return to what it was before pregnancy. There is a high burden on the mother’s circulatory system due to reaching the gestational point that will allow the birth of a child who has a chance of survival, but it is the perinatal period in which we have to deal with the highest mortality rates. It is also important to remember the long-term consequences and risks of PAH progression, as well as any consequences for the child resulting from premature birth and being born with a low birth weight. The available literature data on successful pregnancies in patients with well-controlled PAH do not provide sufficient evidence to recommend pregnancy in such patients [73]. Despite the optimism that should accompany medical staff in the fight for the health and life of a patient, the wide range of targeted therapy in PAH and good effects of exercise rehabilitation training [74], we should also be very critical of the possibilities offered by medicine and be cautious when making decisions, especially as there may be no turning back.

## Figures and Tables

**Figure 1 ijerph-20-04625-f001:**
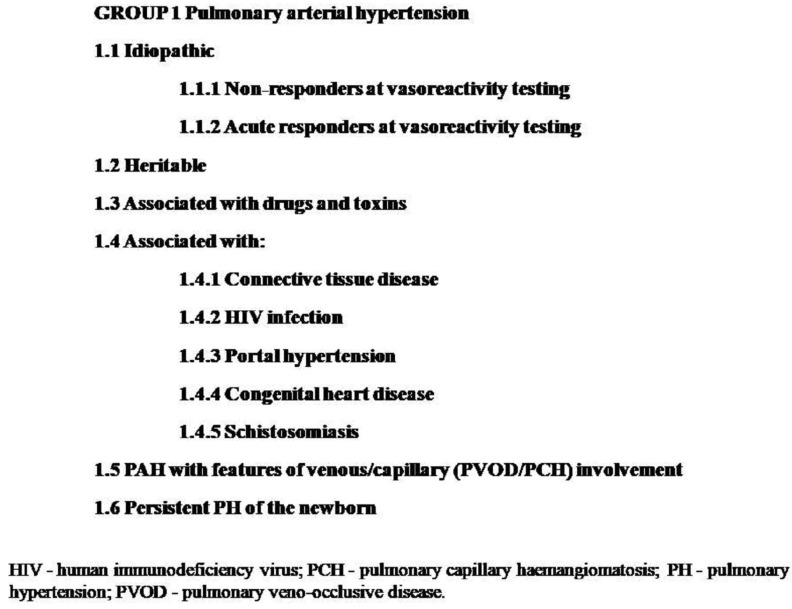
Clinical classification of pulmonary hypertension.

**Figure 2 ijerph-20-04625-f002:**
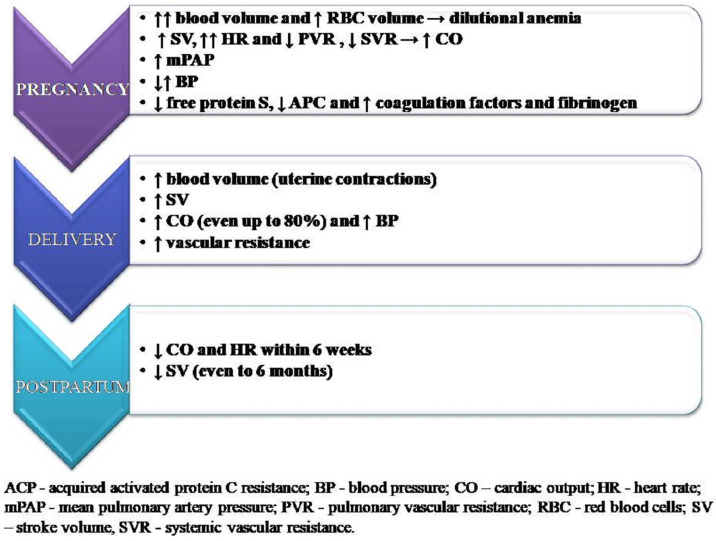
Key hemodynamics changes in pregnancy.

**Table 1 ijerph-20-04625-t001:** Summary of FDA recommendations for the use of conventional therapy for pulmonary arterial hypertension during pregnancy and lactation.

Medication Category	Generic Name	FDA Pregnancy Category	Recommendation during Pregnancy	Recommendation during Lactation
Calcium channelblockers	Amlodipine	C	No adequate studies in pregnant women. Benefit vs. risk.	Unknown presence in human milk. Breastfeeding not recommended.
	Felodipine	C	No adequate studies in pregnant women. Benefit vs. risk.	Unknown presence in human milk. Benefit vs. risk
	Nifedipine	C	No adequate studies in pregnant women. Benefit vs. risk.	Present in breast milk. Benefit vs. risk.
	Diltiazem	C	No adequate studies in pregnant women. Benefit vs. risk.	Present in breast milk, probably in approximate serum levels. Breastfeeding not recommended.
PDE5i	Sildenafil	B	No adequate studies in pregnant women. Not indicated for use.	Present in human milk.
	Tadalafil	B	No adequate studies in pregnant women. Not indicated for use.	Unknown presence in human milk. Not indicated for use.
ERA	Ambrisentan	X	Contraindicated.	Unknown presence in human milk. Breastfeeding not recommended.
	Bosentan	X	Contraindicated.	Unknown presence in human milk. Breastfeeding not recommended.
	Macitentan	X	Contraindicated.	Unknown presence in human milk. Breastfeeding not recommended.
Prostacyclin	Iloprost	C	Benefit vs. risk.	Unknown presence in human milk. Breastfeeding not recommended.
	Epoprostenol	B	No adequate studies in pregnant women. Used only if clearly needed.	Unknown presence in human milk, use with caution.
	Treprostinil	Parental—B	No adequate studies in pregnant women.	Unknown presence in human milk.
		Oral—C	Benefit vs. risk.	Breastfeeding not recommended.
		Inhaled—B	No adequate studies in pregnant women.	Unknown presence in human milk, use with caution.
Prostacyclinanalogue	Selexipag	Not assigned	Limited data.	Unknown presence in human milk. Breastfeeding not recommended.
Guanylatecyclasestimulator	Riociguat	X	Contraindicated.	Unknown presence in human milk. Breastfeeding not recommended.

ERA = endothelin receptor antagonist; FDA = Food and Drug Administration; PDE5i = phosphodiesterase 5 inhibitor.

## Data Availability

Readers can access data supporting the conclusions of the study upon reasonable request to the authors.

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
