# Peer review of "Pregnancy in Patients with Pulmonary Arterial Hypertension in Light of New ESC Guidelines on Pulmonary Hypertension"

_ijerph, 2023, doi:10.3390/ijerph20054625_

Round 1

Reviewer 1 Report

This is an interesting and very informative article „Pregnancy in Patients with Pulmonary Arterial Hypertension in the Light of the New ESC Guidelines on Pulmonary Hypertension

In the introduction, we include detailed descriptions that introduce the subject of the work. In the additional part of the article there are risk factors, birth control, termination, hemodynamic changes in pregnancy, hemodynamic changes during childbirth, pharmacotherapy during pregnancy, labor and the postpartum period, the experience of our center.

They would also like to point out other reports related to the subject of the work from recent years that are related to the subject of the work and may increase the value of the article.

1.     https://doi.org/10.3390/jcm11236932

1.     Did the authors think about introducing a therapeutic - physiotherapeutic program in pregnant patients with PAH?

2.     Please check the correctness of the references and correct them according to the requirements of the journal

Author Response

Comment 1

In our center we haven’t had patients with PAH who were pregnant, and in the available literature there are no data on rehabilitation in pregnant patients with PAH. It is, however, an important addition to pharmacotherapy that allows you to improve health condition of all patients with PAH.

Comment 2

Language has been corrected as suggested

Reviewer 2 Report

General comment

Pregnancy in PAH patients is a risk not only for the mother but also for the newborns. Pulmonary arterial hypertension belongs also to the group IV according to modified World Health Organization (mWHO) classification of maternal cardiovascular risk, what is related to extremely high risk of maternal mortality or severe morbidity. Despite significant improvement in the treatment of patients with PAH, pregnancy is still regarded as a contraindication not only due to high maternal morbidity and mortality but also serious fetal consequences.

Specific points

1- In abstract: the scientific names should be stated in the extended version the first time and then you can use the abbreviation . e.g. WU: Wood units

……………………and pulmonary vascular resistance (PVR) >2 WU.  Wood units

2- I think, the authors should write “pulmonary hypertension crisis” instead of “pulmonary hypertension crises” in introduction section.

The main causes of death, according to the order of occurrence, are right ventricle failure, cardiac arrest, pulmonary hypertension crises, preeclampsia and sepsis.[4,5]

3- I think, the following sentence is too long.

In the case of PAH patients, not only during pregnancy, it is optimal to achieve and  maintain the lowest possible risk profile, in accordance with the current ESC recommendations, no clinical signs of right heart failure, no progression of symptoms, no syncope,  I or II WHO functional class, 6-minute walking distance > 440m, peak oxygen consumption (VO2) > 15 ml/min/ kg, ventilatory equivalents for carbon dioxide (VE/VCO2 slope)  < 36, concentration of brain natriuretic peptide (BNP) < 50 ng/L or N-terminal pro-brain  natriuretic peptide (NT-proBNP) < 300 ng/L, right atrium (RA) area < 18 cm2, no pericardial effusion, tricuspid annular plane systolic excursion/systolic pulmonary arterial pressure index (TAPSE/sPAP) > 0.32 mm/mmHg, right atrial pressure (RAP) < 8 mmHg, cardiac index (CI) ≥ 2.5 L/min/m2, mixed venous oxygen saturation (SvO2) > 65%, stroke  volume index (SVI) > 38 mL/m2 and in MRI parameters right ventricular ejection fraction  (RVEF) > 54%, SVI > 40 mL/m2 and right ventricular end-systolic volume index (RVESVI)  >42 mL/m2 (1) In addition to generally recognized risk factors the first pregnancy, general  anesthesia, and elevated PVR (but without clear cut-off point) were identified as risk factors for the mother.[6,7]

4- I think the authors should review the following sentence (there are some typos).

The heart rate (HR) increases during pregnancy as well as left ventricle end-diastolic diemension and consequently stroke volume (SV) and CI. [20]

5- I think, the authors should not use abbreviation at the beginning of a sentence (e.g. CO, SV).

-CO and HR return to pre-pregnancy values within 6 weeks, but in most cases the recovery is much faster.

-SV also decreases during the postpartum period, but further reduction is seen up to six months after delivery.[32]

6- I think the authors should review the following sentence (e.g. macitetnta)

-Endothelin-receptor antagonists (ambrisentan, bosentan and macitetnta) and guanylate cyclase stimulator (riociguat) are contraindicated in pregnancy due to its teratogenic  properties.

7- I think the authors should review the following sentence (e.g. pospartum)

Pospartum all PAH-specific drugs can be used, regardless of their teratogenic properties.

8- I think the authors should review the following sentence (e.g. determied)

-The first  NT-proBNP concentration determied in 2008 was 182 pg/ml, with small fluctuations and  progression over years to 471,30 pg/ml in 2022.

8- I think the authors should review the following sentence (e.g. permament)

-Sildenafil was introduced to permament treatment and its beneficial hemodynamic effect was confirmed after 4 weeks, the value of PA pressure in RHC was 67.5 / 28.4 (mean 38.4 mmHg) and PVR 736 dyna.

Author Response

Review Report 2

Thank you very much for the valuable comments.

Comment 1

The sentence has been corrected:

.... and pulmonary vascular resistance (PVR) of>2 Wood units (WU).

Comment 2

The sentence has been corrected:

The main causes of death, according to the order of occurrence, are right ventricle failure, cardiac arrest, pulmonary hypertension crisis, preeclampsia, and sepsis[4,5].

Comment 3

The sentence has been corrected

In the case of PAH patients, not only during pregnancy, it is optimal to achieve and maintain the lowest possible risk profile. In accordance with the current ESC recommendations, the determinants of good PAH control include: no clinical signs of right heart failure, no progression of symptoms, no syncope, a I or II WHO functional class, and 6-minute walking distance of >440m. Among the parameters of cardiopulmonary exercise testing the low risk of death is associated with peak oxygen consumption (VO2) of >15 ml/min/ kg, ventilatory equivalents for carbon dioxide (VE/VCO2 slope) of <36, and among the laboratory parameters with the concentration of brain natriuretic peptide (BNP) of <50 ng/L or N-terminal pro-brain natriuretic peptide (NT-proBNP) of <300 ng/L. The echocardiographic parameters of a good prognosis are right atrium (RA) area of <18 cm2, no pericardial effusion, tricuspid annular plane systolic excursion/systolic pulmonary arterial pressure index (TAPSE/sPAP) of >0.32 mm/mmHg, and among the MRI parameters belong here right ventricular ejection fraction (RVEF) of >54%, SVI > 40 mL/m2 and right ventricular end-systolic volume index (RVESVI) of >42 mL/m2. The hemodynamic markers of low risk of death are right atrial pressure (RAP) of <8 mmHg, cardiac index (CI) of ≥2.5 L/min/m2, mixed venous  oxygen saturation (SvO2) of >65%, stroke volume index (SVI) of >38 mL/m2 (1). In addition to generally recognized risk factors, a woman’s first pregnancy, general anesthesia, and elevated PVR (but without a clear cut-off point) have been identified as risk factors for the mother[6,7].

Comment 4

The sentence has been corrected

Heart rate (HR) increases during pregnancy as well as the left ventricle end-diastolic dimension and consequently stroke volume (SV) and CI[20].

Comment 5

The sentences has been corrected

Cardiac output and HR return to pre-pregnancy values within 6 weeks, but in most cases,recovery is much faster. Stroke volume also decreases during the postpartum period, but a further reduction is seen up to six months after delivery.[32]

Comment 6

The sentence has been corrected

Endothelinreceptor antagonists (ambrisentan, bosentan, and macitentan) and guanylate cyclase stimulators (riociguat) are contraindicated in pregnancy due to their teratogenic properties.

Comment 7

The sentence has been corrected

After delivery, all PAH-specific drugs can be used, regardless of their teratogenic properties.

Comment 8

The sentence has been corrected

The first NT-proBNP concentration determined in 2008 was 182 pg/ml, with small fluctuations and progression over the years to 471,30 pg/ml in 2022.

Comment 8

The sentence has been corrected

Sildenafil was introduced as the permanent treatment and its beneficial hemodynamic effect was confirmed after 4 weeks; the value of PA pressure in RHC was 67.5/28.4 (mean 38.4 mmHg) and the PVR was 736 dyna.